



# Research challenges and needs for the deployment of wind energy in atmospherically complex locations

Andrew Clifton[1], Sarah Barber[2], Alexander Stökl[3], Helmut Frank[4], and Timo Karlsson[5]

[1,*]Stuttgart Wind Energy, University of Stuttgart, Stuttgart, Germany
[2,*]Eastern Switzerland University of Applied Sciences, Oberseestrasse 10, CH-8640 Rapperswil, Switzerland
[3]Energiewerkstatt e.V., Heiligenstatt 24, A-5211 Friedburg, Austria
[4]Deutscher Wetterdienst, FE13, Frankfurter Str. 135, 63067 Offenbach, Germany
[5]Tekniikantie 21 Espoo, P.O. Box 1000, 02044 VTT, Finland
[*]These authors contributed equally to this work.

**Correspondence:** Andrew Clifton (andy.clifton@enviconnect.de)

**Abstract.** The continuing transition to renewable energy will require more wind turbines to be installed and operated in many new locations on land as well as offshore. The need to have geographic diversity, as well as limited availability of land in historically "good" locations for wind energy, means that wind turbines will also need to be deployed in hilly or mountainous regions, often known as "complex terrain". These areas can also experience challenging weather and climate conditions and may experience instrument- and blade icing that can further impact their operation. This paper – a collaboration between several IEA Wind Tasks and research groups based in mountainous countries – sets out the research and development needed to improve the financial competitiveness and ease of integration of wind energy in hilly or mountainous regions and in regions subject to icing. The focus of the paper is on the interaction between the atmosphere, terrain, land cover, and wind turbines, and covers all stages of a project lifecycle. The key needs include collaborative research and development facilities, improved wind and weather models that can cope with mountainous terrain, frameworks for sharing data and a common, quantitative definition of site complexity.

## 1 Introduction

The global installed capacity of wind turbines has increased by more than than 10% year-on-year for the last decade. Of the 743 GW of installed wind energy capacity at the end of 2020, around 95% was on land, while the rest is offshore (Lee et al., 2021).

Until the early 2000s, wind energy development generally took place at sites in flat and windy terrain (e.g., Denmark, northern Germany, parts of California and the wind corridor in the Midwestern United States and Canada, and the plains of China). Such areas continue to be popular for many reasons including the wind resource and the relative ease of transportation, installation, and planning.





Regional and global wind resource mapping exercises have shown that significant wind resources can be found in hilly or mountainous locations. Around 30% of the global land surface can be considered to be mountainous (see e.g. Sayre et al., 2018, and Figure 1).

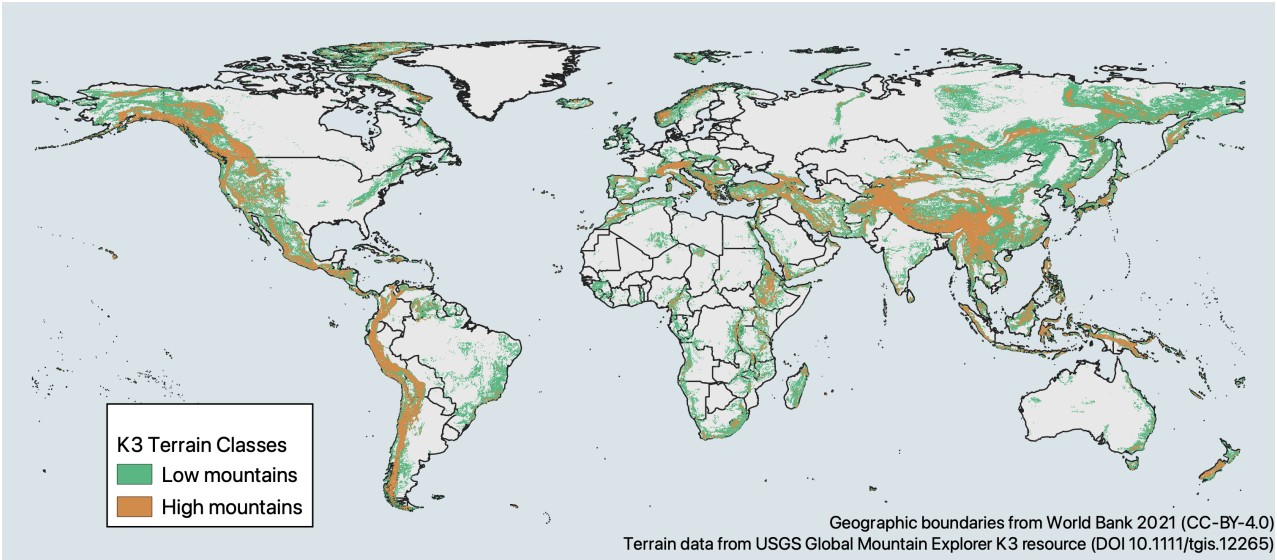

**Figure 1.** The global distribution of mountains.

The ongoing transition to low-carbon energy sources in almost all nations means that such sites are likely to need to be developed as well as flatter sites to meet the increasing demand and to avoid transmission congestion. Also, some regions or

nations simply do not possess sufficient areas of flat terrain and as a result have to build in hilly or mountainous locations if they wish to develop low-carbon energy supplies.

### 1.1    The challenges related to hilly or mountainous locations

Experience shows that despite the high wind speeds that can be found at hilly or mountainous locations, the conditions often bring additional challenges for wind energy development, compared to flat terrain at lower elevations. For example, they can

bring complex flow phenomena such as flow separation, high turbulence, and sudden changes in wind speed and direction that can be difficult to model and also negatively impact turbine performance. They can also bring difficult weather conditions such as storms or icing that make it difficult to forecast plant performance, and can prevent site operations. And, such locations might be more costly or difficult to develop because of steep slopes, narrow roads or poor infrastructure, and snow.

Although a range of flow, meteorological and access conditions all contribute to the challenges of developing and operating

a wind plant in such areas, they are often described simply as "complex terrain" sites. The use of such a broad term makes it difficult to be precise about the challenges and potential solutions and associated research and development (R&D) needs. Instead, in this paper we refer deliberately to different aspects of complexity as follows:





- **Complex flow** refers to wind conditions that cannot be described by simple heuristic wind models such as logarithmic or power law profiles with sufficient accuracy for wind energy applications. Complex flow is not well described by simple heuristic wind models because it exhibits unusual profiles, curvature, or is inclined. These flow characteristics may be caused by winds flowing through a mountain pass or over a ridge or other features, but may also form in flat terrain or offshore. Highly-sheared flows can also be introduced by coastal winds or the diurnal cycle, for example in the Midwestern United States where low-level jets occur over flat terrain (Vanderwende et al., 2015), in Northern Germany (Emeis, 2014), or over the North Sea (Wagner et al., 2019). As well as this, complex flow can also be found as wakes that form behind obstacles such as wind turbines, buildings, islands or mountains. Complex flow might also arise over flat terrain because of variation in surface roughness, leading to the development of internal boundary layers. Such complex flows usually show increased turbulence compared to winds over a smooth, flat, and uniform surface and also may have a different turbulence spectrum compared to idealised boundary layer turbulence. Using the above definition, complex flow may be reasonably expected to occur over mountainous or hilly terrain, within 30 km of coasts (where sea breezes act), and in forested regions. Therefore, aspects of complex flow could well be very common in wind energy development sites worldwide.

- **Complex terrain** is terrain that leads to complex flow conditions, or that leads to other challenges when developing or operating a wind farm. Terrain can lead to complex flow because of its orography through effects such as deflection, detached flows, compression, and channelling. This is often associated with mountains or hilly regions but can also be caused by blockages or gaps.

- **Complex weather and climate** are meteorological conditions such as snowfall, icing, lightning, and storms that impact the operation of a wind farm. It could also include phenomena such as sea breezes, or thermally-driven valley- and slope wind systems, or outflow jets. These may be driven by local terrain (e.g. through elevation) but may also simply be an effect of location, with locations at higher latitudes being more prone to low temperatures. Complex weather is not unusual; a recent IEA Wind Task 19 expert group market forecast showed that around 22% of current wind energy development sites are in "cold climate" locations. That corresponds to more than 150 GW of capacity, or 78,000 turbines (Karlsson, 2021).

In this paper we refer to sites that might have one or more of these characteristics as "complex sites". These characteristics can increase the cost and uncertainty of measurements, are difficult to predict, may lead to high variability between nearby locations (Lange et al., 2017), and can lead to increased wind turbine costs and reduced lifetime. Even the perception of risk may result in increased financing costs, adding to the overall project costs. As a result of these challenges, these types of sites are often avoided by developers.

## 1.2 Mitigating the challenges of hilly, mountainous or forested locations

In order to fully exploit the potential of wind energy, the increased costs and uncertainties resulting from the particular challenges related to hilly, mountainous or forested locations must be overcome. Mitigating these challenges through R&D is



therefore not just of academic interest or relevant for a few developers or wind plant owners, but essential for a successful transition towards a sustainable energy sector.

This paper aims to support this mitigation process by considering the R&D required at each stage of wind energy development, focusing on the interaction of terrain, wind turbine, wind farm and atmosphere. Section 2 looks at the challenges of site prospecting, section 3 at the challenges of resource assessment, section 4 at the challenges of project planning, section 5 at the challenges of wind turbine design, section 6 at the challenges of operational wind plants and section 7 at general challenges. The conclusions are discussed in section 8.

## 2    Site prospecting

Site prospecting is a desktop exercise that uses existing sources of information, such as wind resources, electrical transmission, and infrastructure to identify a potential area for a wind energy project.

At this stage of a wind energy the emphasis is usually on filtering out a large number of candidate areas rapidly and cheaply to allow the next stages of development to progress. As a result of the need for speed and to keep costs down, the tools that are used in this stage are often based on Geographical Information Systems (GIS) and might use data with coarse spatial or temporal resolution. However, such simplified data can be highly inaccurate in mountainous terrain or complex flow. The main challenges in this phase are: (1) Low accuracy of global or national wind data sets; (2) Low availability of local GIS data; (3) Lack of information about the risk of icing. These challenges and the resulting R&D needs are discussed in the next sections.

### 2.1    Low accuracy of global or national wind data sets

Site prospecting often starts by using wind data from global or national data sets to identify sites with attractive wind conditions. These data sets are often known as wind atlases and include the Global Wind Atlas, New European Wind Atlas (NEWA), the Swiss Wind Atlas (SWA), and others (see review in Clifton et al., 2018). There are also a wide range of commercial products available. Because each country is at different stages in the national or regional deployment of wind energy and thus has different needs from an atlas, nationally-sponsored atlases can differ significantly. For example, it was common in the early days of wind energy deployment to simply plot the annual average wind speed and use this for prospecting (see Clifton et al., 2018). Such maps are still often used to communicate the opportunity for wind energy. However, as wind energy adoption increases, there is a need for time-resolved atlases, while in complex terrain higher spatial resolution and increased representation of physical processes such as buoyancy-driven flows is required to capture local effects. Without such steps, atlases can under estimate the wind resources in complex terrain, hide potentially beneficial regional or seasonal correlations, and thus hide development opportunities.

Although many atlases are based on national research efforts, local-scale wind resource data can also be obtained from reanalysis data (e.g., COSMO-REA6 as described in Kaspar et al., 2020). These time-series data products use physics-based modelling, but still lack the spatial resolution required to accurately model winds in complex terrain. Several studies have shown the potential effect of using time-series data from high-spatial and temporal resolution models, compared to time-



averaged wind speed distributions from coarser models. To illustrate the effects for this study we created synthetic time series for one year. One data set has no diurnal variation in wind speed. The other data set has a realistic diurnal cycle added for 50

consecutive days. The diurnal cycles are based on data collected in a Swiss alpine valley (Schmid et al., 2020). A comparison of a 48-hour subset of the data is shown in Figure 2. The difference in mean wind speed between the two data sets is about 4%. The wind speed data was then used to estimate the potential annual energy production (AEP) of a 2 MW wind turbine using a generic, realistic power curve. The AEP increased by more than 20% from 1.5 GWh to 1.8 GWh when the diurnal cycles were added. Although the effect of diurnal cycles will depend on the mean wind speed, the magnitude of the cycle, the turbine's

power curve and other factors, this example shows the need to use highly-resolved data when making development decisions at all stages of the planning process. The 20% difference in AEP seen in this example could easily be significant enough to trigger government interest in supporting wind energy development in a region, or make the difference for a commercial developer deciding between further developing a site or not.

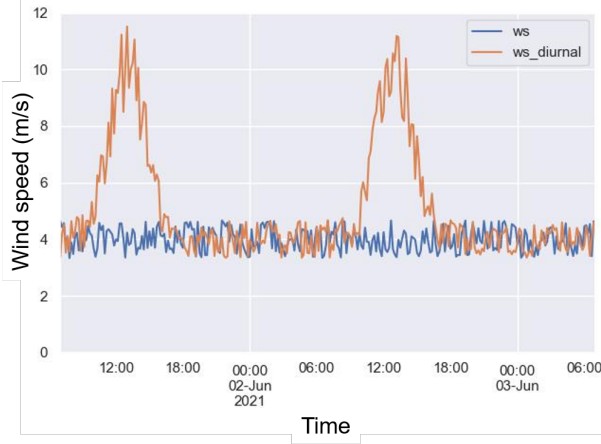

**Figure 2.** Examples of wind time series with similar mean values, with (in orange) and without (in blue) diurnal cycles

The accuracy of global or national wind data sets can be improved by taking into account local weather patterns. They

have been shown to be simulated well using Large Eddy Simulations (LES) (e.g., Chow et al., 2006). However, LES requires prohibitively large computational power as well as high quality input data from weather stations, and it is not feasible to apply the technique to entire regions (such as the Swiss Alps). For other applications, methods for maintaining the high accuracy of simulations but reducing the computational costs have included the Lattice Boltzmann Method (LBM) and machine learning approaches. LBM is a suitable model class for massively parallel simulations of such type of flows (turbulent, thermal, complex

terrain, weakly compressible) with real time potential (e.g., Onodera et al., 2021). The separation of scales of participating phenomena also allows a partitioned approach if necessary (Feng et al., 2021). Some initial studies on applying LBM to wind flow modelling have been done recently (Schubiger et al., 2020); however, there are still a number of difficulties to overcome before it can be used effectively (including high Reynolds number and wall function challenges). Machine learning methods are being used increasingly for extrapolating wind fields (e.g., Foresti et al., 2011), for post-processing weather forecasts (Rasp





and Lerch, 2018) but not yet for efficiently generalising the flow simulation results and projecting the key simulated flow features to all alpine valleys. Therefore, new methods for the extremely computationally efficient prediction of local weather effects at complex sites are required in order to significantly increase the accuracy of wind energy potential estimations.

## 2.2 Low availability of local GIS data

As well as using wind atlases in the prospecting phase, local GIS data is required for planning wind farms. Data on land coverage, zone plan, the grid connection, and slope steepness is key for planning potential wind turbine locations and therefore for AEP predictions. Because these characteristics can change rapidly over small distances in complex terrain it is important to have access to this data at high spatial resolution, compared to flatter locations. This high resolution data is typically not available from governments or agencies and usually has to be purchased directly from third parties.

The low availability of local GIS data can be addressed by developing data marketplaces to help find data, and digital tools that allow easier and standardised access. This should be integrated into the new framework discussed in Section 7.2.

## 2.3 Lack of information about the risk of icing

Wind turbine blade- and instrument icing can potentially result in reduced turbine availability, or reduce the energy production of an operating turbine. This can reduce the attractiveness of a potential development site, but may also penalise sites if an overly-conservative prediction is used. It is therefore important to have accurate but low-cost icing models during the site prospecting phase to correctly account for the potential effects of icing on the wind turbines themselves. The VTT icing map (Figure 3) is one example of how this data can be condensed and made accessible for developers.

Site screening specifically needs simple tools to determine icing conditions from existing measurements, and that can determine the existence of ice throw risk when the icing conditions and potential turbine locations are known. These tools need to be accessible and simple to use without need for complex and detailed simulations.

## 3 Resource assessment

After the wind project site has been chosen in the site prospecting phase, the wind resource has to be assessed in more detail. This involves firstly measuring the wind potential over at least a year and then extrapolating this in time to cover the length of the planned operating period (usually 20 years). After this, the wind field is extrapolated horizontally and vertically in order to cover the entire planned site. Complex sites pose several challenges in the resource assessment phase, including (1) Difficulty planning measurement campaigns; (2) Unknown instrument uncertainty and bias; (3) The sensitivity of remote sensing devices to flow inhomogeneities; (4) Choosing the right measurement instrument; (5) Higher demands on wind field modelling tools; (6) Difficulties in predicting future wind climates. These challenges and the resulting R&D needs are discussed in the next sections.

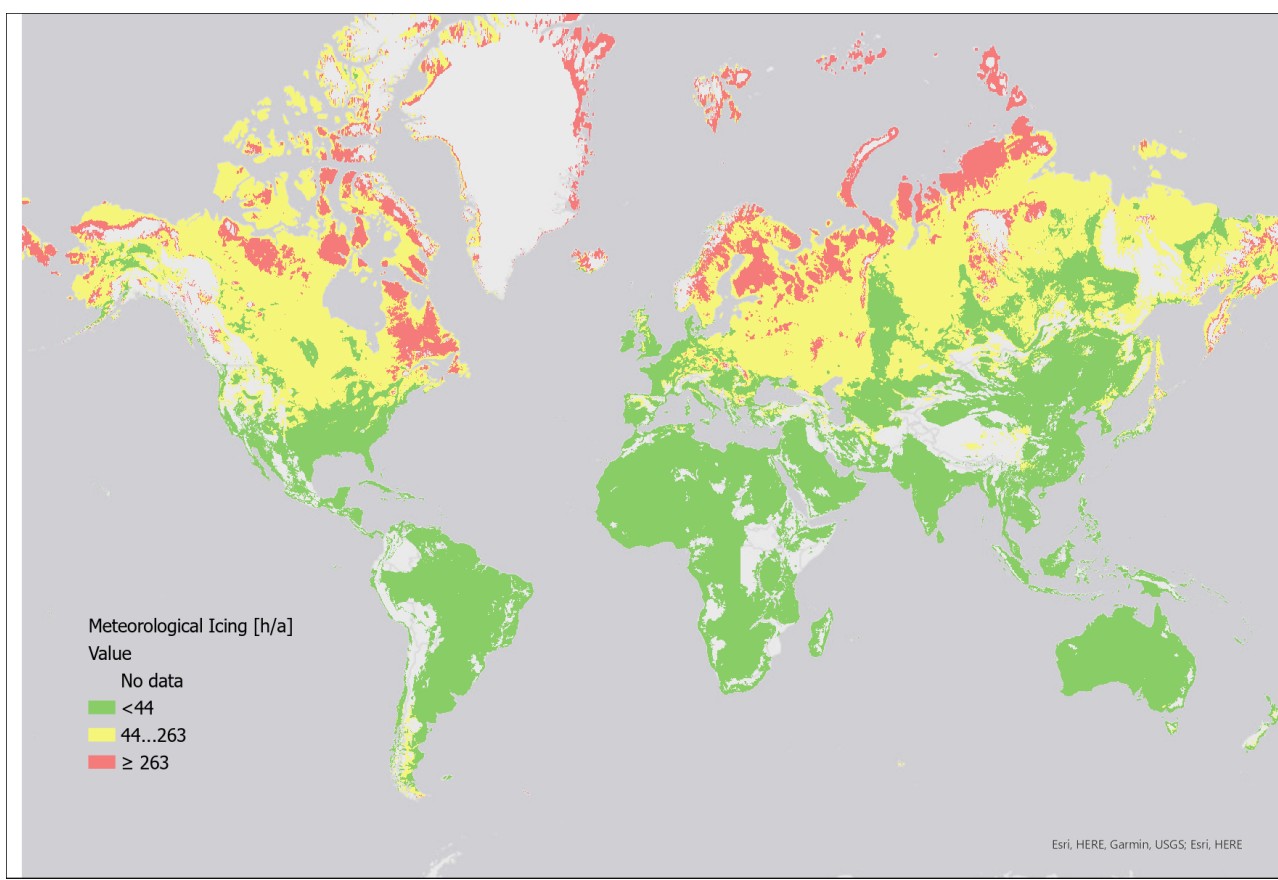

**Figure 3.** The VTT Wind Power Icing Atlas (WIceAtlas), showing the number of hours per year of meteorological icing. An interactive version of the map is available at http://virtual.vtt.fi/virtual/wiceatla/.

## 3.1 Lack of guidelines and planning tools

Accurate and reliable on-site wind measurements are the basis for any wind energy development. They are required to assess the wind resources and ensure the suitability of wind turbine models. When not available from previous experience, it is therefore necessary to gather this information through on-site measurements. The wind energy industry has gained most of its experience with wind measurements in simple terrain and this has been captured in industry guidelines such as the Fördergesellschaft Windenergie (FGW's) Technical Guidelines for Determination of wind potential and energy yields (TG 6). These and other

documents make it relatively easy to design and execute a resource assessment campaign. They also reduce the time taken to plan and initiate a measurement campaign, as well as increase confidence in the results. However, there is a relative paucity of applicable, open knowledge about wind measurements for wind farms in complex terrain, and no applicable standards or guidelines.





Guidelines and experience can be embedded in planning software so that the campaign can be optimised to reduce cost,
uncertainty, or meet some other goal. Some progress has been made towards this for wind lidar deployments with the Campaign
Planning Tool (Vasiljević et al., 2020), but this is limited to measurements using scanning wind lidar. There is therefore a clear
need first for guidelines and standards for resource assessment in complex terrain that can then be used as the basis for other
campaign planning tools.

### 3.2   Unknown instrument uncertainty and bias

The response of measurement instruments can be different in complex flow conditions compared to simple flow. For example,
cup anemometers are by far the most commonly used devices for measuring wind speed. However, they suffer from increased
uncertainty in inclined or highly-turbulent flows (Papadopoulos et al., 2001), and so standards have long limited their use to
a narrow range of inflow angles (e.g., Friis Pedersen, 2003). Three-dimensional sonic anemometers are designed to work
with such complex flows, but historically it has been harder to analyse the data coming from these devices and they tend to
be more expensive than cup anemometers. They are also less reliable in rain or freezing conditions, but can be modified to
work effectively. Similarly, remote sensing devices may need to use data processing approaches that can account for flow
heterogeneity (Bradley et al., 2015). Furthermore, almost all measurement devices can become coated with ice, which can
modify their readings or prevent them from working at all (Swytink-Binnema et al., 2019). These factors can increase the
uncertainty of the measurements themselves as well as the uncertainty of the uncertainty predictions and bias estimates used
to estimate project risks.

There is a need for tools to reliably predict the measurement uncertainty and biases. These should ideally build upon the
methods used for optimising the measurement campaign, but apply on-site measurements as well as expected wind roses and
data from GIS systems. Such tools could also leverage the tools used for wind data extrapolation (see following).

### 3.3   The difficulty of using remote sensing devices to replace met masts

Remote sensing of wind using wind lidar is well established for wind resource measurements, power performance testing, and
site monitoring in simple terrain and offshore. It is clear that the flexibility and ease of use of wind lidar as well as relatively
small size make it ideal for use in complex sites as well, and offer advantages over meteorological towers. However, remote
sensing devices, wind lidar and sodar, use much larger measurement volumes than point measurement devices. This can mean
that they measure in inhomogeneous flows (Figure 4), which in turn may introduce errors in the windfield reconstruction pro-
cess if not accounted for (see e.g. Klaas and Emeis, 2021). Complex flow conditions thus may introduce additional uncertainties
to measurements with remote sensing devices.

Processes have been developed and tested to equate data from volumetric measurements made by profiling wind lidar to
familiar point measurements made by meteorological towers. Experience so far suggests that wind lidar can be used with
confidence in locations that conform to the manufacturer's guidelines (Black et al., 2020, see e.g.), but it is clear that future
wind energy developments may move in terrain or flow conditions that are outside of these existing guidelines.



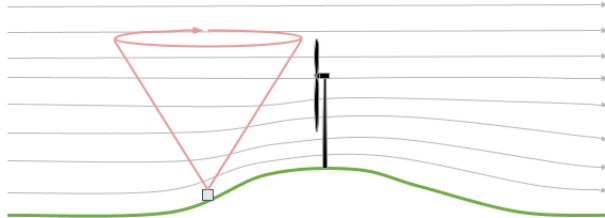

**Figure 4.** Complex terrain can introduce flow inhomogeneity in the measurement volumes of remote sensing devices. In this illustration the terrain (green) causes local speed-up and inhomogenous flow (represented by grey streamlines) through a lidar's measurement volume (represented by the red cone), upwind of a wind turbine.

Research is therefore ongoing into the use of vertical-profiling wind lidar in more complex locations (see e.g., Clifton et al., 2015; Klaas and Emeis, 2021). And, scanning lidar have been seen to be vary useful for measuring wind conditions in complex terrain in research projects (Vasiljević et al., 2017; Menke et al., 2020). However, the experience reported there suggests that such three-dimensional wind scanners must currently be considered research devices that require extensive monitoring and post-processing to obtain usable data. This is unfortunate, given their tremendous measurement capabilities compared to fixed masts. Research is therefore needed into ways to simplify the use of scanning lidar, as well as to process the results. Lidar manufacturers may also need to reduce the cost, weight, and power requirements of their devices to make them easier to deploy.

To truly replace meteorological masts, wind lidar would also have to be able to deliver reliable wind turbulence information. At this time there is no clear consensus about the ability of wind lidar to do this, or the steps that should be taken to equate wind lidar turbulence to the turbulence derived from a cup anemometer (see e.g. Sathe et al., 2011; Newman and Clifton, 2017; Hofsäß et al., 2018). Although there are many possibilities to post process wind lidar data to retrieve turbulence information, the lack of open data sets prevents these from being tested. We therefore suggest that there is a need for a collection of open data sets consisting of colocated wind lidar and anemometers in well-described locations that could be used to validate data processing methods.

The variation in wind energy sites mean that it is unlikely that one type of lidar and one processing approach will work for all sites. But, customised solutions are expensive. Instead we expect to see the development of flexible, digital, modular processes that allow appropriate solutions at each step of the process. This has been seen elsewhere in the wind energy industry and is part of the trend towards greater digitalisation. Research is needed therefore into tools that make it easier to use wind lidar in complex flows, complex terrain, and complex weather. These should leverage available data frameworks such as the e-wind Lidar data format () to build ad-hoc modular processes.

### 3.4 Integrating airborne measurement systems

Although wind lidar partially mitigate the challenges of using meteorological masts in complex terrain, they do not allow high spatial resolution measurements. In contrast, measurement systems on unpiloted aerial vehicles (UAVs) including fixed-wing



aircraft (Rautenberg et al., 2019), helicopters (Hofsäß et al., 2019), and multirotor drones (Molter and Cheng, 2020) can all be used to measure wind vectors and turbulence, as well as other parameters such as air pressure, temperature, and humidity. However, such systems can usually only measure for short periods of time and only measure at one location.

Research is needed into ways to fly multiple systems simultaneously and autonomously (sometimes known as 'swarms'), and to combine the data from these airborne systems with other data sources. This process, known as 'sensor fusion', 'data
fusion', or 'data assimilation' (when used as an input to models) is frequently used in weather forecasting but has not been an active area of research for wind resource assessment.

### 3.5   Choosing the right measurement instrument

The choice of optimal wind measurement device as well as its location and the measurement time period is a critical part of a wind resource assessment and site operations, and is especially challenging for complex sites due to the increased uncertainties
as described above. There is currently no existing guideline, standard or tool available to project planners for doing this. Therefore the development of guidelines, standard or tools for choosing the right measurement instrument would be required to address this challenge.

### 3.6   Higher demands on wind field modelling tools

Wind fields across planned wind farm sites are typically generated using a combination of on-site measurements with some
form of flow model. The modelling is used to extrapolate from on-site data from a few locations that might only extend to a limited height above ground, to the tip of the potential wind turbines across the whole site. This is often known as horizontal and vertical extrapolation.

There is broad academic consensus that the linear flow models that are often used in "simple" terrain simply cannot predict the winds and weather at complex sites because of the steep and changing slopes, forestry, and the effects of atmospheric
stability. There is hope, however, that models that include additional physics may allow to capture effects such as buoyancy or forest canopy effects (see e.g., Knaus et al., 2017; Letzgus et al., 2018). These models are often described collectively as Computational Fluid Dynamics (CFD) models. In turn, there are many different types and fidelities of CFD models, ranging from Reynolds-Averaged Navier Stokes (RANS) models, Detached Eddy Simulation (DES) models, Large Eddy Simulation (LES) models, and Lattice Boltzmann Method (LBM) models (Schubiger et al., 2020). where LES models are often used
in the wind energy community for time-resolved simulations of complex turbulent flows as they offer the ability to resolve turbine-scale flows in realistic terrain (see e.g., Breton et al., 2017).

However, even high-fidelity LES models cannot overcome the greater difficulty of simulating the real flow. As an illustration Figure 5 shows the Root Mean Square Error (RMSE) of 10 m wind speed of the numerical weather prediction model ICON-D2 in March 2021 for sites below 100 m a.s.l., i.e. for rather flat terrain, and for sites above 800m a.s.l., i.e. in hilly and
mountainous terrain. The RMSE is only calculated for sites which are accepted by the assimilation system. The wind speed at sites at higher elevations (i.e hilly and mountainous terrain) is less well predicted than sites at lower elevation.





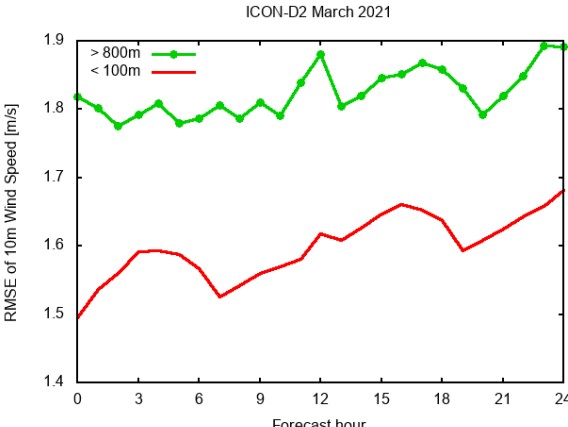

**Figure 5.** RMSE of 10 m wind speed from the numerical weather prediction model ICON-D2 in March 2021 compared to observations at sites below 100m (red) and above 800 m a.s.l. (green). The ICON-D2 domain covers central Europe including most of France to Poland, and from northern Italy to Denmark and thus locations above 800 m are typically in hilly or mountainous terrain. For details of the ICON-D2 model, see Reinert et al. (2021).

These more complex flow models need realistic boundary conditions to deliver accurate results. These may include pressure gradients, surface temperature and moisture conditions, solar radiation or surface heat fluxes, upwind wind profiles, forestry parameters, and other data (Bechmann, 2017). Often these models have many "tuning" parameters, and it is not clear if one

set which was successfully used in one case study is equally good for all weather situations at a different place. As a result, complex models are harder to use than simpler models, both in terms of the data required and the knowledge required to assess the results.

Additionally, there is often no clear evaluation data available for such models, and therefore it is difficult for wind resource engineers to decide on the most effective model for a given site. Although software developers often provide site-specific case

studies, the lack of a clear definition of complexity and an applicable, relevant comparison metric means that it is difficult to transfer experience from one site to another.

The main research need resulting from the challenge of modelling wind fields at complex sites is to improve atmospheric models. Major improvements for wind energy modelling can be expected from better boundary layer schemes and turbulence models. Schemes for the surface layer are often based on Monin-Obukhov theory, which is strictly valid only for homogeneous

sites. Also, in turbulence schemes for atmospheric models horizontal gradients of fluxes and other second order moments are usually neglected compared to vertical gradients. It is not clear at which resolution in complex terrain this simplification is no longer valid. Direct numerical simulation of turbulence is too costly for wind energy assessment.

All models have tuning constants with values obtained in comparing model results to experiments. However, care must be taken not to deteriorate model results when changing them (Sandu et al., 2013). Especially for weather prediction models





there can be conflicting interests. One quantity might improve, but, another one might deteriorate. Hence, changing established values must be done carefully though it might be beneficial in complex terrain.

     Even at mesh sizes of only 2 or 3 kilometers, the subgrid-scale orographic drag must still be parameterized (Olson et al., 2019). The tuning constants of a sub grid scale scheme depend on the ratio of resolved to unresolved orography which depends on the resolution of the original data used to produce input fields for a subgrid scale scheme. This can be critical for the

simulation of winds at typical hub heights.

     Accurate numerical schemes are always critical in complex terrain. Especially the calculation of horizontal pressure gradients should not yield spurious circulations in terrain-following coordinates (Zängl, 2012).

     A second research need resulting from the challenge of modelling wind fields at complex sites is to develop a decision tool for the optimal choice of wind modelling tool. There is a clear need for software or services that uses consistent rules to set up

and run such models, hiding the complexity from the user and thus making it easier for users to adopt them (see e.g., WindNinja [Wagenbrenner et al. (2016)] or WAsP CFD [Bechmann (2017)]). Furthermore, rules- or process-based modelling would give data consumers confidence that the tools have been used appropriately. Recent work involved the development of a decision tool for the optimal choice of WRA tool for a given project at complex sites (Barber et al., 2020a, b, 2021). However, in order to fully develop an effective decision tool, a much larger set of data related to different site complexities, model set-ups and

costs is required. The required skill upgrade, cost of data, and the lack of evaluation data all act as barriers to the adoption of more advanced wind modelling tools.

     Finally, wind modelling needs to follow repeatable, auditable processes that provide the end user with confidence that the results are trustworthy and based on experience gathered at other sites, rather than each site being an independent study. This will require the wind energy industry develop software and services to consistently set up wind flow models. This can be

combined with the data sharing framework discussed in Section 7.2.

### 3.7   Difficulties in predicting future wind climates

An important part of a wind resource assessment is predicting the future wind climates. This estimate may need to extend up to 30 years in the future. It is typically carried out by comparing site data to some kind of long-term reference weather data, such as observations from a nearby automated weather station or reanalysis models, and using this to predict the future wind

resource. This process is known as measure-correlate-predict (MCP) and takes many forms.

     Errors depend mainly on the length of the measurements, and the correspondence to the long time series at the reference site. Both sites must have a similar wind climate, e.g. a coastal station with frequent sea breezes can hardly be correlated with an inland mountain site.

     Such extrapolations depend upon the regional climate in the future being comparable to the past. However, it is not clear if

this will be true as climate change occurs. On complex sites this also concerns changes in long-term and seasonal snow cover in alpine and mountainous regions, which may affect surface temperatures and the associated valley wind systems that contribute to wind energy in some areas. Climate change may also result in changes to the frequency and intensity of storm systems and icing and extreme weather events. While many of the effects of climate change are negative, some may also have positive





consequences for wind energy; it is, e.g., possible that turbines at higher elevations might experience less icing in future than
previously, raising their energy production.

Although some attempts have been made to predict the effects of climate change on wind climates, results suggest that
these effects may be strongly localised and site-specific and as such, cannot be captured using today's relatively coarse global
climate models (Pryor et al., 2020). Although regional climate models offer higher spatial resolution, they may be out of reach
of developers and the wind industry because of the specialist knowledge required to use them.

The first research need related to improving the prediction of future wind climates at complex sites are affordable MCP
processes that can account for the complex wind situations and climates found at complex sites. They need to be reasonably
easy to use so that they can be applied quickly to different locations as part of the site design process. They also need to be
validated using existing sites.

Furthermore, it will be important to develop MCP processes that include the effect of climate change on complex sites.
Because the effects of climate change in mountainous regions are potentially highly localised and site-specific, this might not
be an automated process initially and could instead take the form of expert opinions for sites, identifying risks associated with
different weather conditions due to climate change. In any case, there are no industry-standard approaches to assessing the
impact of climate change on wind energy developments at this time.

The second research need related to improving the prediction of future wind climates at complex sites is to improve methods
for estimating the effect of climate change on wind farm performance. (Pryor et al., 2020)

## 4 Project planning

Following the resource assessment, an energy yield assessment is carried out by firstly using the results of the resource assess-
ment to choose turbines' locations, and then estimating AEP. This information is then used for financial and risk estimations.
Finally, public acceptance for the project has to be gained for it to proceed. The specific challenges related to these steps include
(1) Increased uncertainty of wind turbine performance models; (2) Site-specific wind farm design; (3) Increased financial un-
certainties; (4) Increased conflict potential between stakeholders. These challenges and the resulting R&D needs are discussed
in the next sections.

### 4.1 Increased uncertainty of wind turbine performance models

The power produced by a wind turbine is a function of wind speed, air density, turbulence intensity, shear, veer and many
other factors. Although it is common to apply a site density correction to the power curve according to IEC 61400-12-1 (IEC
International Electrotechnical Commission, 2017), the other factors are difficult to account for, and therefore less frequently
considered. Studies have shown that such atmospheric conditions can affect the turbine output by 10% or more at the same wind
speed, leading to a significant uncertainty in power prediction even if the wind speed and density are known (e.g., Antoniou
et al., 2009; Hedevang, 2014; Vanderwende and Lundquist, 2012; Wagner et al., 2011; Wharton and Lundquist, 2012; Clifton
et al., 2014; Barber and Nordborg, 2020).





Tools that can predict performance at specific sites are therefore required. As well as wind speed, these need to take into account other atmospheric conditions at a turbine's location – such as shear, veer, and turbulence intensity – to estimate the power output at that location. They could be based on experience and leverage data sets from existing power performance tests to generate binned statistics (as explored by the Power Curve Working Group in Lee et al., 2020), or use physics-
based approaches as in the IEC 61400-12-1 standard (IEC International Electrotechnical Commission, 2017). Physics-based approaches have the advantage of being repeatable and easily understood by people, but may not make the best use of the large amount of data available to the wind energy industry.

In contrast, machine learning has the potential to account for the effects of unknown or hard-to-model physics by using power performance data sets to train turbine performance models. These trained models can be used in place of power curves
or physics-based models. Studies suggest that power predictions by machine learning tools trained on wind speed, turbulence, and other atmospheric parameters can reduce the error compared to simple power curves (Clifton et al., 2013; Barber and Nordborg, 2020). The application of machine learning methods to real measurement data is on-going (see e.g., Barber et al., in review). It may be possible to leverage data from power performance tests across a fleet to make more accurate machine learning models. However, machine learning approaches suffer from being "black boxes" in that it is often impossible for a
human to understand what they contain. This can make it hard to include them in a turbine supply agreement or a warranty, for example.

Collaboration between research and industry is required to develop and test more complex power performance prediction tools that use multiple parameters or machine learning, and mitigate the barriers to their adoption.

### 4.2 Site-specific wind farm design

Wind farms are usually designed with the goal of minimising the long-term cost of energy from the site, usually termed the Levelised Cost Of Electricity (LCOE). This is done by optimising the number, size, and layout of turbines on site to maximise energy production and minimise operating costs (Clifton et al., 2016). Accurate wind field models (§3.6) and long-term wind climate data (§3.7) are essential to this; they are foundational for the process of estimating turbine energy production or long-term viability.

Wind turbine energy production at a site is a function of the energy that can be harvested by a turbine, and the losses from that turbine. Wind resource data can be used to predict the energy available from a turbine using power curves (with and without adjustment for turbulence, shear, and veer) or aeroelastic models (e.g. NREL's FAST and others), while other models are required to predict the wakes from those turbines and their impact on downwind machines. It is also important to account for losses due to environmental effects such as blade soiling, and the formation of ice on the turbine blades or instrumentation.
Knock-on effects such as turbine shutdowns to minimise ice throw, or slower maintenance in challenging weather should also be included in the plant energy yield assessment process.

Current wind turbine performance models are designed around inflow angles, shear, and turbulence that lie within standard ranges (defined in e.g., the IEC 61400-12-1 standard). Although there has been some effort to develop power curves that cover a wider range of conditions, these have not been widely adopted or tested openly for complex sites (§5.3). Wakes from turbines





have been extensively measured in simple terrain on land, and offshore. However, there are many fewer wake measurements from more complex terrain, where it is possible that increased turbulence and inclined flows may lead to faster dissipation (Menke et al., 2018).

The challenge is therefore to provide wind farm designers with the information that they need to optimise a wind plant at a complex site. This includes appropriate wind fields and an icing climatology for the location, turbine performance models

that can account for non-standard operating conditions, and wake models that capture the effect of complex flow and terrain on wakes.

Many different wake models exist and many have been validated for use in simple terrain. However, it is not clear how well these models perform in complex terrain or in complex weather situations. Validated wake models would allow increased confidence in energy yield analysis carried out in complex terrain locations. Wake models could be validated through field

measurements, for example combining data from from met masts, wind lidars and wind turbines (Menke et al., 2018). This data would also allow the creation of new wake models. These improved wake models could be used to give better predictions of the wind resources available to downwind turbines.

### 4.3 Increased financial uncertainties

All of the previous factors lead to uncertainty in the potential income from a planned wind energy project.

Electricity from wind energy is usually sold through long-term energy supply contracts with a customer. If the contracts are too expensive, the wind farm owner risks being underbid by another supplier. Therefore, the developer is under pressure to drive the cost of energy as low as possible. However, if these contracts are too cheap (i.e. energy is sold at less than the cost to produce it), the owner risks losing money. To protect against such risks, the project financiers can increase the interest rates on any loans, which in turn increases the project cost and the LCOE.

Project developers typically mitigate these risks by carrying out extensive and detailed pre-construction studies. While these may be more expensive at complex sites than are required in simple terrain, they can reduce the uncertainty enough to reduce the overall project costs and thus justify the extra expense, especially if the site has a high capacity factor. However, there are no guidelines or standards for doing this.

In order to approach the challenge of planning and financing with uncertainties, a guideline for dealing with additional risk

related to complex sites is recommended. This would allow project developers to mitigate the risks by carrying out extensive and detailed pre-construction studies in a standardised and agreed-upon way.

### 4.4 Increased conflict potential between stakeholders

Developing and operating wind energy projects involves a large number of stakeholders. As well as those directly involved with the development, they affect local residents, visitors, and people further away through visual impact, shadow flicker, sound,

traffic, and other mechanisms.

The acceptance of wind farms by stakeholders is one of the major barriers to the adoption of wind energy. Acceptance must be considered for all wind farm developments, both on land and offshore. Experience suggests that wind farm acceptance can



be increased through appropriate and sympathetic wind farm visual and acoustic design (Hübner et al., 2019), coupled with positive stakeholder engagement (Pohl et al., 2018). These challenges may become harder at complex sites because hilly or

mountainous regions may be important for tourism or recreation, wildlife, or other uses, leading to potential conflict between stakeholders (see e.g.,  Straka et al., 2020).

Also, it is possible that the physical processes linked to social acceptance may be harder to predict in complex terrain or at complex sites. Sound propagation from wind turbines is fairly well understood over flat and uniform terrain in uniform wind conditions and can be modelled with some accuracy. In contrast the physical effects of complex terrain or patchy landcover on

sound propagation are less well understood and sound reflection by terrain or damping by forestry have only recently started to be explored (see review in Hansen and Hansen, 2020).

Securing public acceptance is thus one of main challenges the development of wind energy has to face in the next decades. This is part of the growing need to obtain public acceptance – and even more important support – for the far-reaching technological changes connected to the transformation to a carbon-neutral energy generation and the associated social and economic

impact. Developing wind energy in complex terrain is just one focus point where, e.g. the prominent and highly visible siting of wind turbines on peaks and ridges in mountainous regions, may evoke concerns about landscape conservation and touristic and recreational uses. Technical measures such as reducing and managing of the wind turbine's sound and light emissions or changes in turbine design and wind park layout may contribute to a certain degree to the alleviation of these concerns. However, social acceptance of wind energy in complex terrain might also grow from ongoing social transformations through

policy making, fostering of the public understanding of the need for renewables, and the personal participation and benefit from renewable energy projects. One of the initiatives on this interface between technology and social research is the IEA UsersTCP, which also has a big focus on the social acceptance of clean energy technologies.

## 5  Wind turbine design

Complex sites pose challenges for wind turbine design due to the complex flow conditions. This includes (1) Increased im-

portance of quantifying the operating conditions at complex sites; (2) Higher complexity of input conditions for wind turbine modelling; (3) Identifying the freestream wind speed for power performance measurements; (4) Identifying the freestream wind speed for mechanical loads testing; (5) Taking into account icing in the design. These challenges and the resulting R&D needs are discussed in the next sections.

### 5.1  Quantification of operating conditions at complex sites

Operating conditions at sites in complex terrain often fall outside "typical" values. Historically wind energy standards focuson the operating conditions found in development areas such as those found in northern Europe or the American mid west, giving rise to a few standard turbine classes. Small deviations from the operating conditions in such sites are captured using special classes. The design conditions for each special class have to be determined on a case-by-case basis, requiring extra measurements or modeling of the site and extra effort by the turbine OEM, and thus raising costs. And, the lack of understanding



of the operating conditions at complex sites results in a combination of mechanically conservative designs (i.e. with larger safety factors), but may also result in unexpected component failures.

Since then there have been efforts to develop guidelines or standards for wind energy developments in cold climates[1], but in general the trend has been to consider complex terrain to be unique sites and require local measurements of operating conditions as well as extrapolation to the plant life cycle. This is problematic as it add costs for the developer and turbine supplier, and

slows down the development process.

Research is needed to develop tools that can cheaply, accurately, and quickly define operating conditions over complex terrain. These tools also need to account for the effect of forestry and be capable of predicting complex weather associated with the site. This could include realistic time series or spectra of wind resources and complex weather, akin to the standard operating conditions defined in the IEC 61400 family of standards.

## 445   5.2  Wind turbine modelling

Wind turbine design is carried out in the wind energy industry according to the IEC 61400-1 standard (IEC International Electrotechnical Commission, 2019a). In this standard, so-called "load cases" are defined. These refer to particular combinations of external conditions and wind turbine operational status, which have to be considered when simulating wind turbine performance in the design phase.

These simulations are particularly challenging to set up and carry out for turbines at complex sites due to the higher turbulence intensity, shear, veer, temporally varying temperature gradients and extreme changes in wind speed and wind direction.

Validating wind turbine aeroelastic modelling requires accurate, high-resolution information about the inflow to a test turbine, coupled with data from loads and electrical sensors. Meteorological towers that are tall enough are hard to build and operate, while assumptions need to be made about the structure of atmospheric turbulence. Ground-based wind lidar can be

used in some cases, but are not ideal in complex terrain situations.

### 5.3  Power performance testing

Power performance testing according to IEC 61400-12 (IEC International Electrotechnical Commission, 2017) is done as part of the certification process of a new wind turbine type. Power performance testing relates the power produced by a wind turbine to the free-stream wind conditions.

Power performance testing in simple, flat terrain using upwind masts or vertically-profiling remote sensing devices is covered by the IEC 61400-12-1 standard (IEC International Electrotechnical Commission, 2017). This standard specifically excludes winds from directions where there are steep slopes or obstacles from the power performance database. This is because in these conditions it is extremely challenging to identify an appropriate free-stream wind speed, as there may be terrain-induced speed-up or slow-down effects on the flow. As a result, there is no widely-recognised way to perform a power performance test

in complex terrain.

---

[1]Task 19 document



Investigations suggest that it may be possible to fit wind speed measurements made by a nacelle- or spinner-mounted wind lidar looking forward into the turbine's induction zone to a model of the induction, and use this model to estimate the free-stream wind speed (Borraccino et al., 2017). This approach would allow a power curve of power versus free-stream wind speed but has not been widely tested, or standardised.

The recently published IEC 61400-50-3 standard (IEC International Electrotechnical Commission, 2019b) for the use of nacelle-mounted lidar for power performance testing describes the use of wind lidar to measure the turbine inflow wind speed. The wind is required to be measured at more than 2D upwind of the turbine. Modern wind turbines can have rotors with diameter $D$ of more than 150 m and so this could require wind measurements at well over 300 m upwind. However, complex flow conditions could introduce significant flow variation between the measurement point and the turbine (Figure 6), and so it

is not clear that the method can be reliably used in complex terrain.

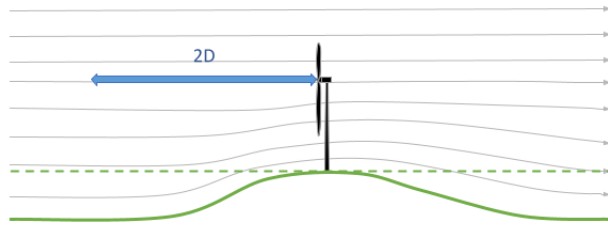

**Figure 6.** Complex terrain can introduce significant flow variation upwind of wind turbines. The 2D arrow indicates the distance two rotor diameters (D) upwind of the turbines.

### 5.4 Mechanical loads testing

The certification process of a new wind turbine type requires mechanical load measurements. These are carried out according to to IEC 61400-13 (IEC International Electrotechnical Commission, 2015). The wind measurements required for this testing are covered by the IEC 61400-12 standard discussed above, and therefore the same challenges apply. Additional challenges to

mechanical loads testing at complex sites relate to the complex behaviour of the loads on the rotor blades due to effects such as shear and veer.

In order to help solve the challenges related to power performance testing and mechanical loads testing at complex sites, field measurements on large wind turbines situated at complex sites are required. This would allow an improved understanding of the actual behaviour of operating wind turbines in the field, enabling OEMs and researchers to improve their design tools

and thus optimise design.

### 5.5 Design for icing

Icing impacts the turbine in several ways and these effects should also be taken into account in turbine design. IEC-61400-1 (IEC International Electrotechnical Commission, 2019a) outlines a number of issues caused by icing that need to be taken into account in turbine design. These include reduced turbine performance due to blade icing, unequal ice distribution on wind



turbine blades leading to unequal loads and increased vibrations, ice shedding from blades, icing effects on wind measurements increased sound levels and prolonged standstills.

These conditions cause issues for turbine control due to ice accretion altering the blade aerodynamics. Turbine control during icing events can have different and competing goals, depending on operator objectives and local regulations. The priority can be chosen to maximise production, to be minimise risks or to minimise additional loads on the turbine components. Additionally,
icing conditions might require extra instrumentation or changes in materials. Active icing mitigation systems such as blade heating will often require changes in turbine design.

An important requirement when designing a turbine to operate in icing conditions is to understand and quantify how ice builds up on the turbine blades, and how this will affect the turbine aerodynamics. This would need to be taken into account when doing simulations during turbine design.

There are existing solutions for these issues. For example, icing on the blades can be mitigated by a blade heating system, anemometers are available on the market that function better in icing conditions, and the risks caused by ice shedding and the issues with increased noise levels can be taken into account when planning the site. The IEA Wind TCP Task 19 report "Available technologies for wind power in cold climates" lists the state-of-the-art solutions that exist in the market (Lehtomäki, 2016). In 2019 Task 19 did a survey on experiences with blade heating and other cold climate solutions and found that many
people working in the field still feel that there is room for improvement in the maturity and reliability of these solutions (Godreau and Tete, 2020). This need for continued testing is part of the reasoning behind establishing the Nergica test centre in Canada, and the RISE cold climate test centre in Sweden.

Many of the solutions for icing need to be designed in to wind turbines and wind plants. For example, safe operation in icing conditions, and optimal blade heating control, will require reliable ice detection. Any ice detection method should be able to
react quickly to icing conditions and also be able to tell when icing conditions and active ice accretion end in order to optimise turbine and blade heating control. In addition, if ice detection is done for safety reasons it's important to be able to tell when blades are ice free.

More detailed icing models are being constantly developed. These models are mainly being validated against wind tunnel measurements (Son and Kim, 2020). Measurements of water droplets during icing events would be very useful. Actual
measurements of ice shapes from an operating turbine are required to validate these models.

There is a large uncertainty related to icing conditions and the icing of turbine air foils. The year-over-year variation of icing conditions can be large and will introduce a large uncertainty in operations. The impact of icing on turbine production also has a large variation that further introduces uncertainty in any estimates on production in an icing climate site. More research is needed to reduce the margin of error in forecasting and modelling production losses (Strauss et al., 2020).

In order to determine the need for icing mitigation, the existence of icing conditions at the site needs to be determined early during site prospecting. The methods for converting these pre-construction measurements into estimates on production losses still have room for improvement. Also, icing conditions need to be known before construction starts in order to determine the need for a blade heating system and the specific operating envelope of a blade heating system (Roberge et al., 2019).





## 6   Operational wind plants

There are a number of challenges associated with operating wind turbines at complex sites, including (1) Lack of standards for performance verification tests; (2) Accurate site-specific power prediction; (3) Forecasting for operational wind farms; (4) Downscaling forecasts to individual turbines; and (5) Predicting icing effects. These challenges and the resulting R&D needs are discussed in the next sections.

### 6.1   Lack of standards for performance verification tests at complex sites

Power performance measurements (see 5.3) are also frequently carried out on newly-commissioned wind turbines or on wind turbines that have been operating for a long period of time. These performance verification tests can reveal problems with the turbine yaw alignment or the turbine control system that can result in several percentage points of lost energy, compared to the optimal setup. However, as with power performance testing during the turbine design phase, the lack of standards for doing this at complex sites makes it very difficult to interpret the results from such tests.

A coordinated research effort and parametric studies on the effect of complex sites on power performance is required. This could involve carrying out a coordinated set of parametric studies using a combination of wind tunnel tests, CFD simulations and field tests at a range of sites with varying complexity and exposed to a range of different complex flow conditions.

### 6.2   Accurate site-specific power prediction

Accurate wind turbine power curves are important for wind farm operation. They can be used for performance monitoring,
calculating compensation for forced curtailment, and for making short-term power forecasts for optimising revenues.

There is therefore a need for site-specific power curves that predict power based on the atmospheric conditions expected at a turbine's location, such as shear, veer, and turbulence intensity. The challenges and needs associated with this are discussed in detail in §5.2.

### 6.3   Forecasting for operational plants

Weather and power forecasting can both increase the income from a wind plant and reduce expenses. They can also enable the integration of wind energy into a regional or national electricity system. As such, effective forecasting is an essential capability for operational wind farms at complex sites.

The ability to forecast conditions at a wind farm several days ahead is essential for forecasting power production, safe operations, and scheduling maintenance. Wind forecasts over shorter horizons can also be used to support plant control decisions,
and in future such insights will be essential for the effective operation of hybrid plants where wind, solar energy, and storage are co-located.

These plant-scale forecasts rely upon understanding the weather in a region up to 1,000 km around the point of interest - i.e., the mesoscale - and predicting it scales of around 1 km or less (the microscale) around the wind farm. These data can then be used directly or further processed, leveraging site observations.





Current generations of mesoscale models are routinely used by commercial and national weather forecasting services at complex sites. However, crucial for a forecast is the model and the initial state or analysis. The analysis is made by a complicated data assimilation process combining short range forecasts and observations. In mountainous terrains the analysis is more difficult because of greater differences in the height of the model orography and the real terrain. Then, simple questions like the observation height become complicated: Should an observation be assimilated at the same height above ground or at the

same height above mean sea level as in nature? Probably, there is no clear answer, and it must be tested for the assimilation system.

Additional forecast errors stem from the inherent uncertainty of the non-linear basic equations of the models. This uncertainty is estimated calculating an ensemble of forecasts. Complex terrain can increase or decrease the uncertainty of wind forecasts by channelling the wind in two preferred directions. This makes the forecast more stable for a wider range of weather regimes.

However, near the tipping point from one regime to the other errors can strongly increase.

Another problem of numerical weather prediction models can be conflicting interests for model improvement. Sandu et al. (2013) showed that reducing turbulent mixing in the ECMWF's IFS model would improve hub height wind speeds, but deteriorate near surface temperature and the large scale circulation.

There are many ways to forecast the amount and timing of energy produced by a wind turbine or wind plant with order (1

minute) resolution up to a week ahead (for a review of approaches, see e.g., Würth et al., 2019). However, every wind plant operator has to go through an evaluation process for their own site when selecting a provider. Although evaluation criteria exist (Möhrlen et al., 2018b, 2019, 2018a), this process is time consuming, requires specialist skills, and the cost of the selection process could be high compared to the savings from the improved forecast.

Simplified and even standardised evaluation processes would help the assessment and adoption of operational forecasts.

Further, sharing anonymised results would help the community and service providers understand where model improvements are required. The challenge is to overcome the wind energy industry's traditional reluctance to share such information. And, objective characterising metrics are also required so that experience can be exchanged (see §7.1).

### 6.4   Downscaling forecasts to individual turbines

Some applications require weather data at individual turbines' locations. This downscaling process can take place using physi-

cal models or by leveraging site observations and applying model output statistics, machine learning, or other methods. Model-based approaches require an understanding of the physics and descriptive equations, while statistical and machine learning tools can be trained on historical data sets. Therefore, the model-based approaches can work better in unusual weather events, but machine learning solutions tend to be faster, and more precise if well trained. However, major changes of the weather prediction model require new training of the statistical model. Otherwise an improvement of the weather prediction model can

result in a degradation of the forecast for the site.

There are many studies about downscaling mesoscale data to e.g., automated weather station locations. But, it is not clear how well such downscaling processes work for wind energy applications, and what the major contributors to uncertainty are. Downscaling - whether by physical models or using statistical approaches - is harder in complex terrain as there is usually





more subgrid scale variation in complex terrain. As all subgrid scale has to be parameterized this introduces greater uncertainty
to the predicted flow. Also, the wind variation is more sensitive to the exact location in complex terrain.

## 6.5  Predicting icing

In icing conditions an icing forecast can also be required in addition to wind forecasts. This icing forecast can be made by
combining a weather forecast model with a wind turbine model to predict ice accretion on the wind turbine blades, or on
the monitoring instruments. An operational icing forecast can have several different use cases. The more common one is to
improve production forecasts. Icing can cause sudden reduction in wind farm energy production. In some electricity markets
this will force the wind farm operator to pay a penalty for missed production. These financial penalties can be avoided if icing
is included in the normal production forecast. Some operators might be concerned about operational safety and ice throw risk
and forecasts can help identify times when there is an elevated ice throw risk.

An operational icing forecast model usually consists of three components: the numerical weather model, a model for blade
ice growth, and an iced turbine model (see examples in Molinder et al., 2020; Kilpatrick et al., 2020). The icing model needs
to take into account not only how the ice builds on the blade, but also how and when ice is removed from the blade.

The ice accretion rate will depend on temperature, wind speed, and droplet size of water droplets in air and turbine specifics
such as blade shape and size. Accurate prediction of the appropriate meteorological parameters and modelling ice accretion
continues to be problematic, but improving them would benefit many different activities in ice-prone regions (Thompson,
2019).

Wind turbine behaviour in icing conditions is specific to a turbine model. Many current models of ice accretion or ice
shedding require detailed information about the turbine, such as the controller design or airfoil shape. Wind turbine OEMs are
often unwilling to share this intellectual property, which in turn prevents the development of operational models. Therefore,
operation models are required that use less sensitive information.

Wind farm operators are typically concerned about the magnitude and duration of icing events. An icing forecast therefore
requires not only an assessment of the meteorological conditions when ice builds on the blades, but also an estimate of how
long ice will remain on the blades and impact turbine performance. The latter is much harder problem to solve as ice can
be removed from turbine blades via mechanical shedding, or by melting, or combination of both. The ice shedding will have
implications on the production forecasts and also on safety around the wind turbines (Andreas Krenn et al., 2018).

As with other issues discussed in this paper, new or improved models of cloud formation, ice accretion, and shedding would
in turn require validation from lab or field tests. It is possible that making multi-scale test data open would allow multiple
different models to be tested and thus accelerate innovation in this field.

## 7  General challenges

The general challenges of complex sites for wind energy applications are those that affect every step of the project life-cycle,
and include: (1) No agreed-upon definition of 'complex terrain'; (2) Interrelated physical processes.





## 7.1 No agreed-upon definition of 'complex terrain'

As discussed in the introduction of this paper, the existing unclear, varying and binary definitions of 'complex terrain' pose a challenge to all parts of the wind farm project life-cycle because it makes it difficult, if not impossible, to quantify the risks associated with project planning and operation as well as to effectively choose optimal measurement instruments, wind models and analysis methods. In this section, therefore, we examine the existing definitions and the important aspects contributing to this challenge.

Modern commercial wind turbines were initially deployed on sites in flat, uniform terrain. This is often described as 'simple terrain' in the wind energy industry. It is easy to recognise by eye; it is characterised by long, flat plains or low-angle slopes, consistent land cover, and a lack of buildings. These conditions allow a deep, uniform wind field to develop. There are several important characteristics of flow over 'simple' terrain:

1. The wind speed profile follows a monotonically increasing, logarithmic profile throughout the atmospheric boundary layer.

2. On the scales typical to a wind farm, any spatial differences in the wind profile are due to pressure gradients. Differences in Coriolis effects can be neglected over such scales.

3. As pressure gradients are low, there are only small differences in wind profiles across a wind farm.

One of the implications of these characteristics is that the wind speed profile across the wind turbine rotor disk (i.e., up to 250 m above ground) can be determined with confidence from measurements made using met masts that might not even reach the hub-height of a future wind turbine.

However, not all flow is 'simple'. Factors including variation in surface conditions, local meteorology, and terrain can all cause wind conditions that deviate from the 'simple flow' case. Following the language used in the wind energy industry, this is implicitly 'complex flow'. However, it is also often called 'complex terrain', leading to difficulties in comparing or exchanging experience. Currently, the catch-all phrase, 'complex terrain', is often used to describe this type of site, although no clear, agreed-upon definition exists. And, the same term is used interchangeably for different applications. For example:

– The software package WAsP is often used in the wind energy industry to model winds at a potential site. The WAsP RiX value (Bowen and Mortensen, 1996) - a measure of the slope angle around a location - has some predictive value for the uncertainty of the linear wind model used in WAsP. Experience shows that WAsP cannot be used confidently when $|\text{RiX}| > 0$, which is then frequently referred to as "complex terrain" in literature. As a result, the widespread availability of WAsP and the simplicity of the RiX metric means that $|\text{RiX}| > 0$ has become the *de facto* definition for "complex terrain".

– Similarly, terrain slope angles are used to assess complexity for power performance testing according to the International Electrotechnical Committee (IEC) 61400-12-1 standard (IEC, 2017). Terrain that exceeds certain angles is said to be "complex".





- The IEC 61400-1 standard (IEC, 2019a) contains a somewhat different scheme that evaluates slope angles and terrain variance and yields a categorisation in low, medium, and high terrain complexity.

WAsP and the IEC 61400 standards therefore refer to 'complex terrain', but use different definitions for it. And, terrain complexity impacts different aspects of the development and operation of wind energy facilities in different ways. Therefore the criteria used to assess complexity for flow modelling using WAsP or power performance testing are not interchangeable, and should not be used for other applications.

    Furthermore, many studies have shown that other modelling tools do not have the same limitations as WAsP (e.g., Tabas
et al., 2019). As a result, it only makes sense to use an absolute value of RiX as a division between "complex" and "not complex" terrain when using WAsP, but not as a general metric for all modelling tools.

    A binary definition of "complex" or "not complex" is difficult to translate into project uncertainties or risks, which are usually assessed on an continuous scale from 0-100% (uncertainties) or at least in 3-4 different categories (risks). This binary definition is in turn difficult to translate into "go" or "no-go" decisions, or to use for deciding which tool or workflow to use for
a particular project.

    As an illustration of these challenges, imagine three wind farm development sites in complex terrain. One is a sparsely wooded and moderately uniform slope, another is a mountain ridge, while another is in a mountain pass (Figure 7).

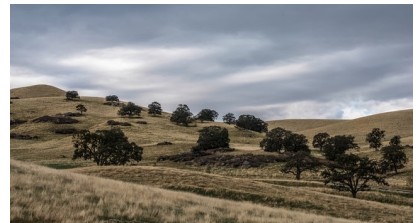 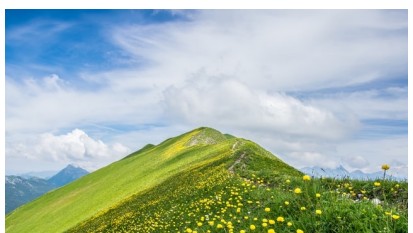 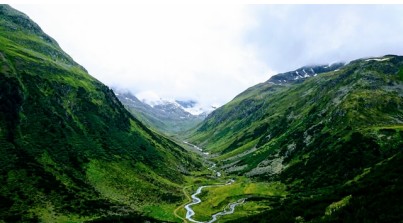

(a) A sparsely wooded, moderately uniform slope      (b) A mountain ridge      (c) A mountain pass

**Figure 7.** Examples of three different types of complex terrain. Images are exemplary only and do not represent potential wind energy development sites. Photo credits: a) Delaney Turner, b) Adrian Jakob, and c) Tomoe Steineck. All photos are from Unsplash.com and used under the Unsplash license.

Experience suggests that each site will have different pre- and post-construction challenges related to the interaction of the terrain and weather:

- A uniform slope is subjectively relatively simple terrain, but is likely to experience a diurnal cycle of up- and down-slope winds, driven by surface heating and cooling. As a result, wind flow models are required that can generate such buoyancy-driven flows.





- A mountain ridge has more apparent complexity because of the marked three-dimensional relief. Accurately predicting the wind resource means accurately predicting the effect of the ridge terrain and landcover on the local wind fields, which in turn requires the hilltop geometry to be resolved in the models that are used.

- Winds in a mountain pass are driven by a combination of thermal effects and regional pressure gradients (Clifton et al., 2014). Accurately resolving the wind resource requires very high resolution modelling that also includes buoyancy.

These examples illustrate that any scheme for assessing site complexity should be able to quantify the different aspects that can lead to the characterisation of a site as a "complex terrain" site. Moreover complexity might not be the same for all applications, methods, and devices.

We therefore consider the lack of a clear and transferable definition (or definitions) of site complexity to be an important challenge for the development of wind energy at complex sites.

## 7.2 Interrelated physical processes

The wind energy industry leverages knowledge from many different scientific disciplines to design, build, and operate a wind farm. Knowledge is transferred between wind energy projects in the form of computer models that approximate the many different physical processes taking place, and their interaction.

For example, predicting the energy that might be produced over the course of a year requires information about the wind resource, the ability of the wind turbine to capture that energy, and losses due to icing, soiling, and other effects. This might be implemented as a complex computer model. The performance of each individual model in the this system can be verified by conducting experiments that isolate specific effects (e.g., the effect of atmospheric stability on wakes, described in Bodini et al., 2017). The performance of the whole modelling system can be verified against field experiments that simultaneously resolve physical processes acting at different scales and in different parts of the system and their effects on wind turbines (e.g., the US Wind Forecasting Improvement Projects WFIP I, (Wilczak et al., 2015) and WFIP2 (Shaw et al., 2019), and the New European Wind Atlas (Mann et al., 2017)). However, these are extremely complex and expensive studies to carry out, and, as wind turbines' dimensions increase and their rated power increases, these experiments need to be repeated.

This is particularly challenging for complex sites due to the additional complexity of the flow and its interaction with the wind turbines.

These challenges could be solved in the following ways:

1. **New frameworks for sharing data:** There are a number of desktop tools available for supporting wind energy site development, which can be used to estimate resources, carry out coarse energy yield assessments, and identify other challenges in advance of significant effort on site. Some tools additionally allow resource estimates based on downscaled mesoscale weather models or reanalysis data that can then be coupled with simple turbine production models. However, there is no framework we know of for bringing together data to rapidly analyse the opportunities and challenges of a potential wind energy development site. This data exchange may be enabled by the ongoing digitalisation of the wind



energy industry. For example, in 2021 IEA Wind Task 43 produced a data structure to allow simplified data sharing for
wind resource assessments (Holleran et al., 2021).

2. **A network of test facilities in complex terrain, weather and climate locations:** To date several wind turbines have
been erected in part or entirely for research purposes in complex terrain, including the Gütsch site in Switzerland and
the CENER Alaiz experimental wind farm in Spain, or in complex flow conditions such as those found at times at
the US National Wind Technology Center near Boulder, Colorado, or in the complex weather and climate found at the
Nergica facility in the Gaspe region of Canada. A further wind energy research facility is in development by the southern
German wind energy research cluster WindForS in the Swabian Alb. These facilities individually cover different parts of
the spectrum of terrain, flow, and weather complexity and include a range of turbines, from 600 kW machines at Gütsch
to multi-megawatt turbines at Alaiz and the NWTC. Despite the ongoing trend towards taller wind turbines with larger
rotors and higher rated power, so far there are no research turbines of greater than 100 m hub height or more than 5 MW
rated power in complex terrain that also experience complex weather and climate. This lack of available research turbines
makes it difficult for the international wind energy community to identify and test solutions to the issues identified in
this paper. A turbine of this scale would potentially cost well over €20M to procure and construct (but operations could
be self-funding through the income from power sales), and so might require an international consortium to realise it.

**8 Conclusions**

Wind turbines have been successfully deployed in all kinds of operating conditions from the Arctic to Antarctic and at up to
4,000 m above sea level. However, wind energy developments in the so-called "complex terrain" found in hilly or mountainous
regions are still viewed negatively despite potentially increased wind resources and the potential of such sites to support the
global transition to low-carbon energy.

The ongoing development of wind energy in complex flow, terrain, weather and climates has shown that wind farms can be
productive and economic, and can contribute to the ongoing energy revolution. Complex sites can have markedly increased
wind resources compared to other locations, making them potentially significant contributions to energy supplies in many
regions.

However, deploying wind turbines at complex sites is not easy. The interaction between winds, terrain, weather and climate
leads to bigger turbine- and wind farm-scale variability than are usually found at traditional deployment locations. Conditions
can be more extreme than at "typical" locations, and it can be hard to obtain realistic or representative data from measurements
or models. Together these lead to costly uncertainty that can make it difficult to finance a new wind energy project. Operating
wind farms in such conditions is not easy, either; weather may make it hard to work, forecasts are often less precise than in flat
terrain, and there may be conflicts with other stakeholders.

Focused R&D is therefore required to maintain the competitiveness of wind energy at complex sites. While some of the
outcomes of this R&D will be able to be applied to existing wind farms, larger benefits can be obtained by designing new wind





farms from the outset with the site's complexity in mind. The development of appropriate metrics to characterise terrain and flow, as well as the tools and processes used in such terrain will be a key enabler for this R&D. It will allow the exchange of experience across developments, and help investors understand the applicability of different tools. Finally, platforms and

frameworks are required to bring together the complex terrain wind energy community, where stakeholders can collect and share their experiences.

In this article we have shown how sites at complex sites may have different challenges compared to simple sites. However, there are challenges to deploying wind energy at complex sites that can occur in simple terrain, such as complex flow, weather and climate conditions. As a result, research and development for wind energy at complex sites can benefit the entire wind

energy industry.





**Table 1.** Challenges for wind energy in hilly, mountainous, or icing-affected locations and the associated R&D needs

| Stage in life cycle | Challenge | R&D need |
|---|---|---|
| Site prospecting (§2) | Low accuracy of wind atlases | Inclusion of local and seasonal weather effects; time series databases; increased spatial resolution |
| | Low availability of local GIS data | Data marketplaces; digital tools for easier data discovery and integration; data sharing frameworks |
| | Lack of information about icing risk | Simple tools to assess icing risk; tools to estimate ice throw risk |
| Resource assessment (§3) | Lack of guidelines and planning tools | Standards for resource assessment in complex terrain; software for planning measurement campaigns built on this guidance |
| | Unknown instrument uncertainty and bias | Tools to predict uncertainty and bias in different flow conditions |
| | The difficulty of using remote sensing devices to replace met masts | Tools for processing wind lidar data collected in very complex situations; simplify the use of scanning lidar; develop modular lidar processes to enable custom data processing toolchains |
| | Integrating airborne measurement systems | Improved vehicles; swarm operation; sensor fusion and data assimilation |
| | Choosing the right sensor | Guidelines or tools for instrument selection |
| | Higher demands of wind field modelling tools | Improved atmospheric models; decision tool for optimum choice of wind modeling tool; repeatable, auditable, experience-based processes |
| | Difficulties in predicting future wind climates | Develop reliable MCP processes for complex sites; improve methods for estimating the effect of climate change on wind farm performance |
| Project planning (§4) | Increased uncertainty of wind turbine performance models | Data sets for verification of multi-variate power performance models; acceptance of black-box approaches |
| | Site-specific wind farm design | Data needed for multi-variate power performance models; wake models for complex terrain |
| | Increased financial uncertainties | Guidelines for dealing with additional risk at complex sites |
| | Increased conflict potential between stakeholders | Better understanding of the sources of stakeholder conflict; better understanding of the physics of sound in complex terrain |
| Wind turbine design (§5) | Quantification of operating conditions at complex sites | Standardised operating conditions for complex sites; tools to cheaply and accurately estimate operating conditions |
| | Wind turbine modelling | Quick desktop tools; high-resolution data for model validation |
| | Power performance testing | Standards for using nacelle-mounted lidar in complex terrain |
| | Mechanical loads testing | Field testing on large turbines in complex terrain |
| | Design for icing | Improved icing models; improved solutions for blade and instrument icing; improved AEP estimation accounting for icing |



**Table 1.** (continued) Challenges for wind energy in hilly, mountainous, or icing-affected locations and the associated R&D needs

| Stage in life cycle | Challenge | R&D need |
|---|---|---|
| Operational wind plants (§6) | Lack of standards for performance verification tests at complex sites | Standards for using nacelle-mounted lidar in complex terrain |
| | Accurate site-specific power prediction | Multi-parameter power prediction tools; use and acceptance of machine learning |
| | Forecasting for operational plants | Downscaling to the plant location or individual turbines; simplified or standardised model evaluation processes |
| | Predicting icing | Improved weather models; improved ice accretion models; improved turbine performance models; climate-controlled test facilities; test facilities in icing locations |
| General (§7) | No agreed-upon definition of "complex terrain" | Clear, transferable definition(s) of complexity that are relevant to the different processes happening there |
| | Interrelated physical processes | New frameworks for sharing data; network of test facilities in complex terrain, weather and climate locations |



*Author contributions.* This paper was initiated and led by AC. AC and SB equally contributed material and edited the document. AS, HF, and TK contributed materials on resource assessment, forecasting, and icing respectively.

*Competing interests.* AC, AS, and TK provide consultancy on the deployment of wind energy in complex terrain. All authors except HF receive grant funding for research into aspects of wind energy in complex locations.

*Acknowledgements.* AC was funded by the Ministerium für Wissenschaft, Forschung und Kunst Baden-Württemberg during the writing of this paper.



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
