# Peer review of "Research challenges and needs for the deployment of wind energy in hilly and mountainous regions"

_Wind Energy Science, 2022_

## Author Comment (AC1)

**WES-2022-11 Reply to reviewers**

June 2022

**1 Reviewer 1**

The following comments were published as https://doi.org/10.5194/wes-2022-11-RC1.

Dear authors,

thank you for a valuable overview of the challenges of complex terrain. Overall I quite liked the paper, and have just a few minor comments.

Two comments to start with: you seem to lump operational forecasting together with wind resource assessment, which is not getting so clear from the introduction. It's fine to discuss both, but it was my feeling from the start that you were mainly discussing WRA.

We think the reviewer is referring to the discussion of "Higher demands on wind field modeling tools" in what was §3.6, now 3.7

The paper actually refers to the different challenges found through the whole lifecycle of the wind farm. We have emphasized this in the abstract, introduction, and elsewhere.

... And is there no other noteworthy issue than icing, since you have that particular effect quite prominent? Lightning or dust, e.g.?

The reviewer is right that there are other potential challenges for wind energy in hilly or mountainous regions. We have noted this in §5.1, where we discuss the lack of understanding of operating conditions at complex sites.

Line 24+: "Also, some regions...have to build in hilly..." The assessment here sounds a bit negative. Often you also see wind power erected on hills to get the speed-up effect on hill tops, which is an active choice.

We have rewritten the introduction to provide a clearer motivation for R&D to support wind energy deployment in hilly or mountainous terrain, and have noted the positive reasons for doing this.

L102: Hopefully, to use more effort has a positive effect, not just a potential effect?

Unfortunately it is often the case with numerical weather prediction tools that adding physics to the model increases the data that is needed for initial or boundary conditions, but this data is not always available. Similarly, the additional physics may not be complete or may be tuned for different conditions. Thus, adding physics can result in more expense for the same (or even worse) results. Therefore we described this in the first version as a 'potential' effect.

Figure 2: Looks weird. What is the high-frequency floor at 4 m/s, why is it not changing at all during the two days, and from the plot there is no way that both time series have the same mean (within 4%). The diurnal "mountain" goes to 10m/s, flattening the steep slopes to a rectangle gives some 4-5 hours duration per day, and the floor is pretty much identical. With those simple geometric assumptions, I get the red line to be over 5m/s in average.

We agree that ths figure was not well explained and did not significantly contrbute to our arguments. We have removed it from the manuscript.

L222: Drones can cover an area, or is that not what you are thinking of here?

Drones can measure only at a single point, but can move. We have modified the text in this section slightly to clarify this and other use cases.

L242: Should you also include Direct Numerical Simulation (DNS) in this list?

The list of flow simulation tools has been removed and replaced with a reference to a summary paper.

L248: ICON-D2 is to my knowledge not a LES code?

The discussion of different types of flow simulation tools has been removed from this section.

The ICON-D2 comparison has been moved to the section on forecasting for operational wind plants.

L306+: There are attempts to make climate models more user friendly under the Copernicus initiative from the EU ("C3S").

Thank you. This has been added to §3.8

L454: On the difficulties of ground-based lidars you probably have a reference?

A reference has been added. This sentence has also been edited for clarity.

L506: please add references to both test centres.

These references have been added. The sentence has been moved to the end of the subsection.

L644: The rationale for the RIX was that from the slope of 30%, the flow is non-attached, and is outside the operational envelope of WAsP. It would be nice to also mention both critical slopes in the comparison.

These are helpful comments but we have been unable to find supporting references in the literature. We have not changed the manuscript.

Typos and other editorial issues (please only answer to those if you disagree):

- Abstract lines 2+3, please check the sentence structure.

  The first three sentences of the abstract have been edited for readability and to match the new introduction.

- L81: "At this stage of a wind energy" - project?

  This text has been modified and the correction is no longer required.

- L142: check sentence structure.

  This text has been modified and the correction is no longer required.

- L187-190: "This can mean that they measure in inhomogeneous flows" sounds maybe not quite right here. And the measurement volumes are only a part of the problem, the way the lidar averages over a circle is also to blame (as you also illustrate in Figure 4).

  The section on the use of wind lidar in complex terrain and flows has been revised for clarity. It is now §4.3.

- L197: vary → very?

  We have removed this word as it is subjective.

- L216: There seems to be a reference missing at "()"?

  Reference added.

- L289: ...require the wind industry *to* develop... ?

  Added.

- L421: IEA Users_TCP ?

  Fixed.

- L430: focus_on

  Fixed.

- Page 17, footnote 1: "Task 19 document" is not a good reference, sounds more like a placeholder during the writing process...

  Fixed.

- L494: to be minimise →to minimise ?

  Sentence modified for clarity.

- L565: ...from one regime to the other, errors... (add comma?)

  Added.

- L705: Task 43 *Digitalisation* (adding the short qualifiers to the task names makes them easier to remember).

  Added.

- L750: The Acknowledgement is quite unspecific, usually they require project numbers.

  This acknowledgement meets the requirements of the funding agency.

- And you often seem to have an extra blank before your references, but maybe that's just some layouting which will go away in the final paper trim. E.g. L90, L100, L173, L333, L339, L406, L600

  This was our error and has been fixed.

**2 Reviewer 2**

The following comments were published as https://doi.org/10.5194/wes-2022-11-RC1.

General comments:

The article sets out a fairly comprehensive set of research challenges for wind energy in "atmospherically complex" locations.

My main feedback is that this article should be seen as adding to similar "challenge" articles in recent years. For example, van Kuik et al (2016) (not cited in the article under review) and Veers et al (2019) (not cited in the article under review). While the two aforementioned published challenges articles go beyond atmospheric science aspects, they do feature many atmospheric challenges. I would like the authors first to recognise these aforementioned challenges papers and state how their article goes beyond what is stated in those publications.

References to van Kuik et al (2016) and Veers et al (2019) have been added in the introduction. We see a need for a self-contained paper that identifies these challenges directly and feel that such a paper would be particularly relevant for researchers and funding agencies based in regions with complex terrain. This is stated now explicitly in §1.3.

I think the article is rather long. I can see the purpose of the article, which really is focussing on complex terrain locations, but I lack a succinct summary and set of recommendations. Perhaps Table 1 can be made more central to the article and have additional columns with headings such as "impact" and "priority" following the R&D column. Then the main text can be reduced and focussed on discussion points around the table.

We agree that the article was somewhat long and have used this revision as an opportunity to trim some content and sharpen our focus. We note that at approximately 18 pages in final version form it is similar in length to other application area reviews recently published in WES, e.g. *Floating wind turbines: marine operations challenges and opportunities* by Ramachandran et al. (2022) DOI: 10.5194/wes-7-903-2022. It will be significantly shorter than the "challenges" papers.

We have also modified the paper to provide summaries of the challenges and needs in a table at the end of each section. These tables include an assessment of the current TRL and the potential impact on both project risk and LCOE.

One last comment, while terrain complexity and icing are mapped (Figures 1 and 3), I think the "complex flow" and "complex weather and climate" are quite overlapping and very general terms. It would be difficult to map these. Where is weather and climate simple, one could ask? Therefore it is almost a corollary; that where there is complexity there is also challenge.

We agree that there was an overlap between "complex flow" and "complex weather and climate". We have revised our definitions to include "complex flow", "complex terrain", and "cold weather and climate". These are included in §1.2.

Specific comments:

Abstract: Please state briefly what is the new conclusion emerging from this review beyond what can be found in earlier "challenges" articles.

As noted above, The various "challenges" articles do not specifically address the challenges for wind energy in complex terrain. We see a need for a self-contained paper that identifies these challenges directly and feel that such a paper would be particularly relevant for researchers and funding agencies based in regions with complex terrain. This is stated now explicitly in §1.3.

L20 What is the definition of mountainous in Fig 1 and main text.

The definition used is complex and so has been added as an appendix. The main text and Figure 1 provide cross references.

L25 Mention offshore resource can also be used if "simple" terrain is exhausted or not available.

As noted in our response to Reviewer 1, the introduction has been rewritten to better explain the motivation behind this paper.

L30 Insert "...and sudden *associated* changes in wind speed..."

This text has been removed.

L57 "Complex weather and climate" seems a bit loose. Is there "simple weather and climate"? See general comment above.

We have revised this definition to focus in on "cold weather and climate".

L70 "relating to hilly, mountainous or forested locations", is that the same as complex terrain discussed on same page? If so, can that term be used?

See comment re. revised introduction. Text has been removed.

L110 Fig 2 I completely agree that the diurnal cycle is important to capture. But this is not a new result. Furthermore time series data is not required to capture that. Knowing variance of wind speed, via for example, $k$ (shape parameter) together with $A$ (scale parameter) of a Weibull distribution of wind speed, would capture this too effect too.

This subsection has been revised. The figure and discussion have been removed and replaced with relevant references.

L195 I think a reference to Bingöl would be fitting here:

Bingöl, F.: Complex terrain and wind lidars, Dissertation, Risoe National Laboratory for Sustainable Energy, Technical 545 University of Denmark, Roskilde, 2009. Bingöl, F., Mann, J., and Foussekis, D.: Conically scanning lidar error in complex terrain, Meteorologische Zeitschrift, 18, 189–195, 2009.

L350 "human"-¿"person" perhaps?

Agreed. Changed.

L357 About the word "maximise" and "minimise" in the sentence. I do not agree with the sentence. Layouts should rather maximise the ratio of production to costs. It is not the same.

This sentence has been revised.

L430 "focuson"-¿"focus on".

Fixed.

L474 Figure 6, it is not completely clear what this figure is illustrating. Is it needed?

We agree that this figure is not needed. Removed.

L545 Please state how expenses can be reduced?

This has been added in §6.3

L562 Consider changing "inherent uncertainty" with "chaotic nature" or similar.

This text has been removed. Was "Additional forecast errors stem from the inherent uncertainty of the non-linear basic equations of the models". Now reads, "Additional forecast errors stem from the non-linear basic equations of the models."

L619 It is not surprising there is no agree-upon definition of "complex terrain" because the definition will probably depend on the use case or model in questions. I would argue that when speaking of complexity, one needs to qualify the measure being used, e.g. what threshold slope value is being used etc.

Agreed. Our point was that there is however effectively a *de facto* definition of complex terrain which is inappropriate for the majority of applications. In §7.1 and in the conclusions we encourage different groups to establish and share appropriate definitions of complexity for their applications.

Several places "RiX"→"RIX", it stands for Ruggedness INdex.

Changed.

L669 "interaction of the terrain and *flow*" is perhaps better in this context.

Agreed. Changed.